# Arboviruses in Free-Ranging Birds and Hematophagous Arthropods (Diptera, Nematocera) from Forest Remnants and Urbanized Areas of an Environmental Protection Area in the Amazon Biome

**DOI:** 10.3390/v14102101

**Published:** 2022-09-22

**Authors:** Bruna Alves Ramos, Liliane Leal Das Chagas, Franko de Arruda e Silva, Eder Barros dos Santos, Jannifer Oliveira Chiang, Joaquim Pinto Nunes Neto, Durval Bertram Rodrigues Vieira, José Wilson Rosa Junior, Eliana Vieira Pinto da Silva, Maria Nazaré Oliveira Freitas, Maissa Maia Santos, Jamilla Augusta de Sousa Pantoja, Ercília de Jesus Gonçalves, Landeson Junior Leopoldino Barros, Sandro Patroca Silva, Carine Fortes Aragão, Ana Cecilia Ribeiro Cruz, Lívia Medeiros Neves Casseb, Lizandra Caroline dos Santos Souto, Joana D’Arc Pereira Mascarenhas, Erilene Cristina Da Silva Furtado, Raimundo Nelson Souza Da Silva, Alexandre do Rosário Casseb, Lívia Carício Martins

**Affiliations:** 1Department of Arbovirology and Hemorrhagic Fevers, Evandro Chagas Institute, BR 316, km 07, s/n, Levilândia, Ananindeua 67030-000, PA, Brazil; 2Department of Virology, Evandro Chagas Institute, BR 316, km 07, s/n, Levilândia, Ananindeua 67030-000, PA, Brazil; 3Central Laboratory of Pará State (LACEN-PA), Boulevard Augusto Montenegro, 524, Guajará Park, Belem 66823-010, PA, Brazil; 4Health and Animal Production Institute, Federal Rural University of Amazonia, Boulevard Pres. Tancredo Neves, 2501, Terra Firme, Belem 66077-830, PA, Brazil

**Keywords:** arboviruses, birds, hematophagous arthropods, urbanization, Amazon

## Abstract

The rapid and disorderly urbanization in the Amazon has resulted in the insertion of forest fragments into cities, causing the circulation of arboviruses, which can involve hematophagous arthropods and free-ranging birds in the transmission cycles in urban environments. This study aimed to evaluate the circulation of arboviruses in free-ranging birds and hematophagous arthropods captured in an Environmental Protection Area in the Belem metropolitan area, Brazil. Birds were captured using mist nets, and hematophagous arthropods were collected using a human protected attraction technique and light traps. The birds’ sera were subjected to a hemagglutination inhibition test to detect antibodies against 29 arbovirus antigens. Arthropod macerates were inoculated into C6/36 and VERO cell cultures to attempt viral isolation and were tested using indirect immunofluorescence, subsequent genetic sequencing and submitted for phylogenetic analysis. Four bird sera were positive for arbovirus, and one batch of *Psorophora ferox* was positive for *Flavivirus* on viral isolation and indirect immunofluorescence. In addition, the *Ilheus virus* was detected in the sequencing and phylogenetic analysis. The presence of antibodies in sera from free-ranging birds and the isolation of *Ilheus virus* in *Psorophora ferox* indicate the circulation of arboviruses in forest remnants in the urban center of Belem.

## 1. Introduction

Arboviruses are viruses naturally maintained in cycles involving hematophagous arthropods and vertebrate hosts [1]. They are endemic in tropical regions, affecting animals and humans throughout the year, due to the biotic and abiotic conditions of these areas, which are favorable for the reproduction of hematophagous arthropods and conducive to coexistence between vectors and vertebrate hosts in the same environment [2,3].

Hematophagous arthropods are arbovirus vectors and reservoirs. After acquiring infection, they remain infected for life, being able to transmit these viral agents to the vertebrate hosts through hematophagy [4]. Several insect species of the Culicidae, Ceratopogonidae, and Psychodidae families are competent arbovirus vectors that are responsible for the transmission of these viruses to humans and animals and the maintenance of sylvatic and urban cycles of the most known arboviruses [5,6,7].

Birds are considered amplifying hosts for some arboviruses. They are the main source of infection for competent arthropod vectors and participate in sylvatic enzootic cycles of arboviruses worldwide [8,9]. Free-ranging species, especially those with migratory behavior, play an important role for arbovirus dissemination, silently transporting these and other infectious agents from endemic to non-endemic areas, thus characterizing themselves as a One Health problem [10].

The environmental pressure exerted by human expansion into preserved environments, with forest fragmentation and the formation of “wild corridors”, promotes the approximation between wild animals, vectors, and humans, allowing the emergence or reemergence of arboviruses of medical and veterinary importance in large urban centers [11,12].

The rapid and disorderly expansion of the large cities, including forest invasion and fragmentation, is a common problem in the Amazon, where approximately 212 arbovirus species have already been isolated, 104 of which are exclusive to the region and 36 of which are associated with human diseases [9,13,14]. Cities such as Belem, the capital of Pará, have been spreading and entering adjacent sylvatic areas, leaving only forest fragments protected by law, such as the Environmental Protection Area of the Metropolitan Region of Belem (APA-Belem) [14,15].

APA-Belem is an Environmental Conservation Area with 74.57 km^2^ of extension located in the municipalities of Belem and Ananindeua, comprising forested areas for leisure, ecotourism, environmental preservation, and teaching and research, such as the university headquarters of the Federal Rural University of the Amazon (UFRA) [15]. Approximately 30 arboviruses unique to the Amazon region were isolated for the first time in the APA-Belem jurisdiction [16,17]. In addition, arboviruses of medical and veterinary importance such as Saint Louis encephalitis virus (SLEV), Eastern equine encephalitis virus (EEEV), Western equine encephalitis virus (WEEV), and Mucambo virus (MUCV), which were isolated from vectors and free-ranging birds inhabiting the site [18].

This study aimed to evaluate the circulation of arboviruses by detecting antibodies in serum samples of wild free-ranging birds and the infection in hematophagous arthropods (Diptera, Nematocera) captured in forest fragments and urbanized environments located in the APA-Belem jurisdiction.

## 2. Materials and Methods

### 2.1. Study Area

The study was conducted at the university campus of UFRA, Avenida Presidente Tancredo Neves, Terra Firme neighborhood, Belem, Para, Brazil, with coordinates 1°27′31″ S 48°26′04.5″ W, and comprising an area of approximately 215,230 ha, within APA-Belem. Thus, contemplating distinct landscapes characterized by the presence of buildings, roads, pastures, plantations, and buildings intended for the creation of domestic production animals and discontinuous forest fragments, composed (mostly) by secondary forest.

### 2.2. Ethical Consent

The procedures described below were approved by the Brazilian Institute for the Environment and Renewable Resources (register number 63488-4 ICMBio-MMA), the Animal Use Ethics Committees (CEUA) of the Federal Rural University of the Amazon (register number 025/2018—CEUA; 23085.01479/2018-82—UFRA), and the Evandro Chagas Institute (register number 16/2019 CEUA/IEC).

### 2.3. Capture and Identification of Free-Ranging Birds and Hematophagous Arthropods

Four 10-day excursions to the campus were conducted to capture free-ranging birds and collect hematophagous arthropods in rainy (February/March), rainy-dry transition (June/July), dry (September/October) 2019, and rainy seasons (March) 2020.

The free-ranging birds were captured using single-height mist nets (2.5 m), measuring 7 to 12 m in length, armed with the aid of metal rods fixed to the ground, and arranged in rows. Four rows were set inside a secondary forest fragment, and three armed in a moderately urbanized environment close to the Veterinary Medicine sector (Figure 1). Capturing started at 6:00 am and ended at 11:00 am, with inspection of the nets every 30 min and immediate removal of trapped animals. After capture, the birds were taxonomically identified in order, family, genus, and species by comparing their physical characteristics with data present in physical field guides [19,20] and bird identification applications [21,22]. The nomenclature recommended by the Brazilian Committee of Ornithological Records [23] was used for taxonomic identification.

Hematophagous arthropods with diurnal habits were collected using a human protected attraction technique [24], whereas those with afternoon/nocturnal habits were collected using CDC light traps. Both techniques were applied at ground level and in the tree canopy at two different points: inside a secondary forest fragment located near the Animal Husbandry sector and inside the same forest fragment where the birds were captured (Figure 1). Daytime activities began at 9:00 am and ended at 12:00 pm. The traps were turned on at 4:00 pm the day before and disarmed at 9:00 am the following day. For the taxonomic identification of individuals collected in family, genus, subgenus, and species, dichotomous keys described by Consoli and Oliveira [13] were used. Only arthropods belonging to the Culicidae, Ceratopogonidae, and Psychodidae families were considered in this study.

### 2.4. Sample Collection

Blood samples corresponding to 1% of the live weight of each free-ranging bird were collected by puncturing the jugular vein to obtain the serum samples. Individuals weighing less than 10 mg were not included in the collection. After collection, blood aliquots were conditioned in individual gel tubes for each bird, left at room temperature, and centrifuged at 2000 rpm/10 min, for serum separation.

To obtain the macerates of hematophagous arthropods, microtubes (lots) containing 1 to 50 individuals were added with a tungsten bead and 1 mL of 1 × D-PBS diluent (Gibco, Waltham, MA, USA) plus 5% of fetal bovine serum, Fungizone and Penicillin/Streptomycin, and the contents were macerated by shaking in a TissueLyser device (QUIAGEN^®^) at 25,000 Hz/1 min. After maceration, the suspensions were stored in a freezer (−80 °C/24 h), thawed at room temperature, and centrifuged (10 rpm/10 min) to obtain the supernatant.

### 2.5. Hemagglutination Inhibition (HI)

Free-ranging bird serum samples were submitted to HI, following the methodology described by Clark and Cassals [25], adapted for microplates by Shope [26], using a 1:20 cut-off. To perform the test, 50 µL of each serum sample was treated with saline solution and acetone P.A, adsorbed on treated erythrocytes of geese (*Anser cinereus*) to remove non-specific antigens, and later tested against 4 A.U of antigens of 29 different arbovirus (Table 1), belonging to the Serology 1 Laboratory of the Department of Arbovirology and Hemorrhagic Fever of the Evandro Chagas Institute (SAARB/IEC). Positive samples were titrated using serial dilutions of bovine albumin up to a maximum titer of 1:1280 to obtain the antibody titer.

### 2.6. Viral Isolation in Cell Culture and Indirect Immunofluorescence Test (IIF)

For viral isolation, 150 µL of the supernatant from each macerated batch of hematophagous arthropods was inoculated simultaneously into appropriate cell culture tubes (TPP^®^) containing a monolayer of *Aedes albopictus* intestinal cells (C6/36 ATCC: CRL 1660 clones) [27] and African green monkey kidney cells (VERO ATCC: CCL 81 clones) [28]. The inoculated cultures were incubated at 28 °C (C6/36) and 37 °C (VERO) for seven consecutive days and observed daily under an inverted field microscope to verify the presence/absence of cytopathic effects. In addition, all inoculated cell cultures were subjected to the IIF test using the protocol adapted from Gubler et al. [29] and tested for eight distinct antigenic groups of arboviruses (Table 2), using polyclonal antibodies belonging to the Viral Isolation Laboratory of the SAARB/IEC.

### 2.7. Nucleotide Sequencing and Sequence Assembly

RNA extraction was performed from the inoculated C6/36 culture supernatant, using the Quiamp^®^ Viral RNA mini kit, with subsequent quantification of the extracted RNA using the Qubit RNA HS Assay kit (Invitrogen^®^) and Qubit 4 Fluorometer equipment, following the manufacturer’s recommendations. Subsequently, the synthesis of the first and second strands of cDNA was performed using SuperScript VILO^TM^ Master Mix and NEBNext mRNA Second Strand Synthesis Module kits, respectively, and the cDNA purification and quantification were performed using the PureLink^TM^ PCR Purification and Qubit DNA HS Assay kits (Invitrogen^®^), following the protocols described by the manufacturers. Genomic libraries were produced using the Nextera XT DNA kit, and sequencing was performed using the NextSeq 500 platform and the NextSeq 500/550 High Output kit v.2.5 (300 cycles), using the paired-end methodology.

To generate the R1 and R2 reads, the sequencer output data were demultiplexed using the bcl2fastq program, and both were evaluated for quality using the FastQC program (https://www.bioinformatics.babraham.ac.uk/projects/fastqc/, accessed on 9 February 2022). The comparison of R1 and R2 readings with databases of non-redundant proteins (nr) was performed using DIAMOND program [30], considering the values of e-value (0.0001) and amino acid identity as parameters. Visualization of tabulated data in DIAMOND was performed using the KRONA v.2.8 program [31]. The raw data were processed using the programs SortMeRNA v.2.1 [32] and Trim Galore v.0.4.5 (https://www.bioinformatics.babraham.ac.uk/projects/trim_galore/, accessed on 9 February 2022), removing interferents such as ribosomal RNA, readings with less than 75 nucleotides, adapters, and indeterminate bases (above 15 N).

The sequences were assembled using the DE NOVO methodology, using SPAdes [33] and IDBA-UD [34] programs, considering k-mer values 21, 33, 55, 77, and 20, 40, 60, 80, and 100, respectively. The obtained contigs were grouped and analyzed using the DIAMOND program (e-value 0.0001), and those that showed similarity with virus sequences were analyzed by gene prediction using the GeneMarkS program [35]. The amino acid sequences of the genes were compared to different protein databases available in the InterProScan program (https://www.ebi.ac.uk/tnterproscan/dowload, accessed on 9 February 2022), and the tabulated contigs were inspected and compared with contigs from complete arbovirus sequences using Geneious software v.9.1.4 (https://geneious.com/, accessed on 9 February 2022).

### 2.8. Phylogenetic Inference

To construct the phylogenetic tree, two partial fragments corresponding to the coding regions of the envelope proteins and NS5 of the nucleotide sequence obtained in the present study, together with partial sequences referring to the same proteins from strains of ILHVs available at the NCBI, were submitted to multiple sequence alignment using the Maft program v.7 [36]. The aligned data were analyzed to identify the best nucleotide substitution model, and the phylogenetic trees were constructed using the maximum likelihood (MV) method [37] in the IQ-TREE v.1.6.12 program [38]. A bootstrap test was applied using 1000 replicates [39] to add greater reliability to the clusters. FigTree v.1.4.4 (https://www.github.com/rambaut/figtree/releases/tag/v.1.4.4, accessed on 9 February 2022) was used to visualize the phylogeny, and the final image was produced using InkScape v.1.1 (https://inkscape.org/release/inkscape-1.1/, accessed on 9 February 2022).

## 3. Results

### 3.1. HI

One hundred and twenty-eight sera from free-ranging birds were subjected to HI test. Four (3.1%) had antibodies to one or more arbovirus antigen tested, with two (1.8%) positive for *Phlebovirus* and two (1.8%) positive to *Flavivirus*. In addition, a sample of *Ramphocelus carbo* showed a monotypic reaction to ICOV, a sample of *Sclateria naevia* showed a monotypic reaction to BUJV, a sample of *Dendroplex picus* showed a monotypic reaction to SLEV, and a sample of *Taraba major* showed a heterotypic reaction to SLEV and NJLV (Table 3).

### 3.2. Viral Isolation

Four hundred and thirty-three batches of hematophagous arthropods were inoculated into C6/36 and VERO cell cultures and subsequently processed using the IIF test. A batch identified as BeAr865640 containing 24 individuals of *Psorophora* (*Janthinosoma*) *ferox*, collected using human attraction protected and enlightened technique at ground level, next to the Animal Husbandry sector, during the dry season, caused cytopathic effect in both strains cells on the fourth day after inoculation, and was positive for *Flavivirus* in the IIF test (Figure 2).

### 3.3. Nucleotide Sequencing and Phylogenetic Inference to ILHV

The supernatant of C6/36 cells inoculated with batch BeAr865640 was subjected to nucleotide sequencing, and a sequence of 10,745 nt was identified, with 5′ and 3′ region representing 78 and 389 nucleotides, respectively, and a coding region containing 10,278 nt, with a high degree of identity with the complete genome of ILHV available from NCBI. The average sequencing genomic coverage was 37,551×. The sequence of the isolated ILHV strain was deposited in the GenBank database under identification number ON553739.

The coding region of isolate BeAr865640 was aligned and compared with the coding regions of three complete ILHV sequences available at NCBI (Figure 3), evidencing the nucleotide and amino acid divergence sites between them. Furthermore, the identity matrix containing the percentage of nucleotide and amino acid similarity between the isolate BeAr865640 and the strains MH932545, KC481679 and NC_009028, showed a high degree of identity between the amino acids produced by the isolate and those produced by the sequences, with a lower degree of nucleotide identity between them (Table 4). This finding may indicate the occurrence of synonymous mutations in the isolate, where there was a change of one or more nucleotides in the codon without changing the amino acid produced.

Regarding the phylogenetic tree, the ILHV strains deposited at the NCBI formed two distinct monophyletic groups in relation to the envelope and NS5 proteins: one group consisting of isolates from Brazil and Central America, and the other formed by strains isolated in Peru, Ecuador, and Brazil. The isolate BeAr865640 was found in a branch external to the two monophyletic groups and was more closely related to the group of isolates from Brazil and Central America (Figure 4).

## 4. Discussion

Birds are natural amplifying hosts of most arboviruses and may harbor approximately 80 viral species that cause encephalitis in humans and domestic animals, such as SLEV, ILHV, WNV, and EEEV [40,41]. At least 31 different arboviruses have been detected in the biological samples of wild free-ranging birds in the Amazon, highlighting the importance of these animals in the arbovirus cycle in the region [40]. The detection of antibodies in sera from wild, free-ranging birds is an important indicator of arbovirus circulation and can be applied to monitoring these agents in preserved natural environments and areas affected by human action. Serological tests, such as HI, are fundamental tools used in arbovirus surveillance, as they detect antibodies produced in the initial stages of infection that last long periods [1].

Hematophagous arthropods are vectors of arboviruses that can be found in different environments and coexist with animals and humans [7,42]. Approximately 300 hematophagous arthropod species can participate in the cycle of urban and wild arboviruses, especially the Culicidae family, which is related to the transmission of most arboviruses [43]. After infection, culicids and other vectors remain infected with arboviruses throughout their lives and are considered reservoirs of these infectious agents in nature [44]. Thus, entomological surveillance and laboratory tests such as viral isolation and molecular tests using hematophagous arthropods are essential in epidemiological research on arboviruses, considering the importance of hematophagous arthropods as maintainers of arboviruses in nature and their potential as transmitters of these diseases to vertebrate hosts [4,42].

In this study, antibodies against ICOV, BUJV, SLEV, and NJLV were detected in sera from wild free-ranging birds, and ILHV was isolated from *Psorophora (Jan.) ferox* vectors captured in preserved forest fragments and urbanized areas, demonstrating the occurrence of wild cycles of arboviruses of importance in One Health withing the APA-Belem.

Previous serological and virological studies have demonstrated the presence of natural cycles of arboviruses that cause encephalitis in APA-Belem, emphasizing the importance of the occurrence of these infectious agents within APA for public and animal health [11,17,18]. For example, a serological survey developed by Bernal et al. [45] involving rodents and marsupials captured in secondary forest fragments located inside the university campus UFRA-Belem detected 57.14% positivity for presence of antibodies to arboviruses of the families *Togaviridae*, *Flaviviridae,* and *Peribunyaviridae*, demonstrating the possible circulation of these viral agents in wild animals living in preserved forest areas present in the APA-Belem. In addition, a study developed by Barros [46] in the Environmental Protection Area of Combu Island (APA-Combu), located close to APA-Belem, detected the presence of HI antibodies for ICOV, BUJV, ILHV, SLEV, NJLV, and other arboviruses in sera collected from wild free-ranging birds, demonstrating the circulation of arboviruses in other environmental protection areas in Belem that are influenced by anthropic action.

*Icoaraci phlebovirus* and BUJV are two arboviruses of the *Phenuiviridae* family isolated for the first time in the 1960s from samples of rodents captured in the Utinga forest, in the territory covered by the APA-Belem [47]. Rodents, non-human primates, and marsupials are considered amplifying hosts in sylvatic cycles of *Phlebovirus* from the New World [48]. Free-ranging birds can be considered accidental hosts of ICOV, and their participation in the cycle of this arbovirus and BUJV is still poorly understood [40,49]. Sandflies are the most important vectors in the transmission of arboviruses of the *Phenuiviridae* family worldwide, together with some species of nocturnal culicids and ceratopogonids of the genus *Culicoides*, which can also participate as vectors in the wild cycles of these arboviruses [50,51]. Phlebovirus infections cause a self-limiting febrile illness in humans, which may present symptoms similar to those of other viruses and are easily confused with other diseases [52].

*Saint louis encephalitis virus* is an arbovirus that causes encephalitis in humans and domestic animals, with an enzootic cycle that involves wild birds and mosquitoes of the genus *Culex*. Is the *Flavivirus* has the highest seroprevalence among free-ranging birds in the Amazon, holding a cycle in which approximately 91 species of birds are involved as amplifying hosts [40]. In addition, secondary cycles of SLEV involving vectors of the genera *Aedes* sp., *Coquillettidia* sp., *Mansonia* sp., *Psorophora* sp., *Sabethes* sp., and vertebrates such as rodents, non-human primates, horses, and humans, have been reported in Brazil [1,53]. Humans are considered terminal accidental hosts of this arbovirus, presenting varied susceptibility to the virus, and may develop mild or severe disease characterized by symptoms such as fever, myalgia, headache, meningitis, and encephalitis [53].

*Naranjal virus* was first isolated from samples from a sentinel hamster in the city of Naranjal, Ecuador [16]. In Brazil, this arbovirus was isolated from a marsupial sample of *Metachirus opossum* species captured in the area of influence of the Salobo Project in Marabá, Pará [54]. Phylogenetic studies have demonstrated that NJLV belongs to the Aroa group and is included in the clade of encephalitogenic flaviviruses transmitted by *Culex sp.*, which is genetically related to BSQV [55,56,57]. Studies by Freitas [58] and Barros [46] mentioned the presence of hemagglutination inhibitor antibodies to NJLV in free-ranging birds captured in the area of influence of the Salobo Project and the APA-Combu. According to Gaunt et al. [55], flaviviruses included in the Culex clade generally present free-ranging birds as the main natural amplifying hosts. Dégallier et al. [40] suggested that BSQV has an enzootic cycle that involves vectors of the *Coquillettidia* and *Sabethes* genera, rodents, non-human primates, and birds. As NJLV has a strong genetic relationship with BSQV, the possibility that free-ranging birds participate in the enzootic cycle of this arbovirus cannot be ruled out, and additional studies are necessary to elucidate the role of these animals in the transmission and maintenance cycles of NJLV.

*Ilheus virus* was first isolated in 1944 in Ilhéus, Bahia, Brazil, from mosquitoes of genera *Aedes* and *Psorophora* [59]. It is an arbovirus with variable morbidity in humans, which can cause asymptomatic infection, mild symptoms, or severe febrile illness accompanied by encephalitis [1,60]. The sylvatic cycle of this arbovirus mainly involves mosquitoes of the *Psorophora ferox* species and free-ranging birds, among other vectors and vertebrate hosts [60,61,62]. *Psorophora ferox* females are opportunistic zoophilic hematophagous that feed on human blood when there is an increase in the population of the species and when humans are present in or around forested areas [13]. Sporadic cases of human encephalitis caused by ILHV have been reported in Central [63,64] and South America [65], including southeastern Brazil [62]. Despite the low isolation rate, the prevalence of antibodies to this arbovirus in the human population ranges from 2 to 30% and may be higher in places with proven viral circulation [66,67].

The environmental impacts of disorderly urbanization, deforestation and fragmentation of pre-existing forests, associated with climate change, put pressure on vectors and wild amplifying vertebrate hosts to adapt to the urban environment, bringing them closer to humans and domestic animals and supporting the emergence of sylvatic arboviruses in large cities in the Amazon [11].

Humans and domestic animals are terminal accidental hosts of most arboviruses and may present asymptomatic infection or develop symptoms similar to those of other diseases with greater occurrence in urban areas, thus making clinical diagnosis difficult [2,52]. In addition, the lack of specific laboratory diagnostic tools for the detection of several arboviruses makes it difficult to detect wild arboviruses in cities because it does not identify the viral species involved in the infection.

Thus, the intensification of epidemiological, entomological, animal, and environmental surveillance, associated with laboratory investigations, using molecular techniques such as nucleotide sequencing and phylogenetics analysis, is a very important tool for monitoring the circulation of wild arboviruses and detection of new viruses in preserved and altered environments located in large Amazonian urban centers.

## 5. Conclusions

The detection of antibodies to encephalitogenic arboviruses in sera from wild free-ranging birds and the isolation of ILHV in *Psorophora ferox* vectors indicate the occurrence of cycles of wild arboviruses in preserved and altered environments in the territory covered by APA-Belem. Such findings are important for One Health, given the importance and location of the APA, which supports contact between vectors, wild vertebrate hosts, domestic animals, and humans, and consequently, the greater risk of emergence of wild arboviruses, not only in the territory corresponding to the APA-Belem, but also in urban areas located in its surroundings.

## Figures and Tables

**Figure 1 viruses-14-02101-f001:**
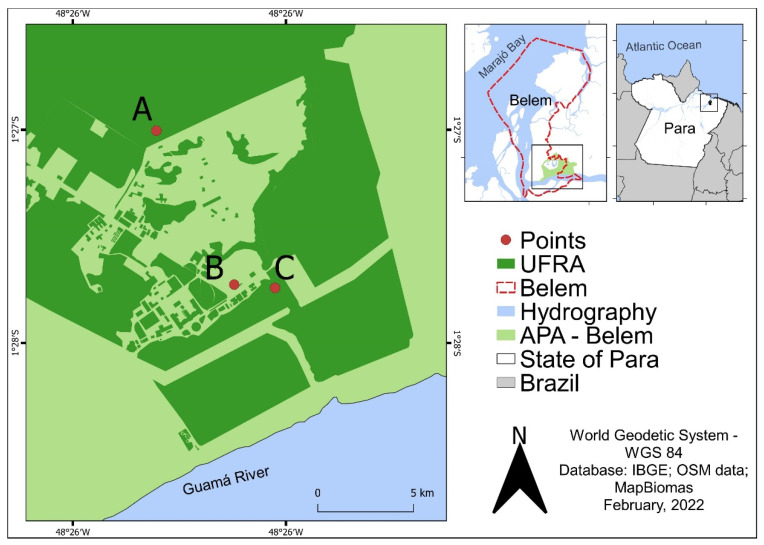
Location of the UFRA inside the APA-Belem (Pará, Brazil), with the demarcation of points for collecting hematophagous arthropods (A and C) and capturing free-ranging birds (B and C). A—Animal Husbandry sector. B and C—Veterinary Medicine sector.

**Figure 2 viruses-14-02101-f002:**
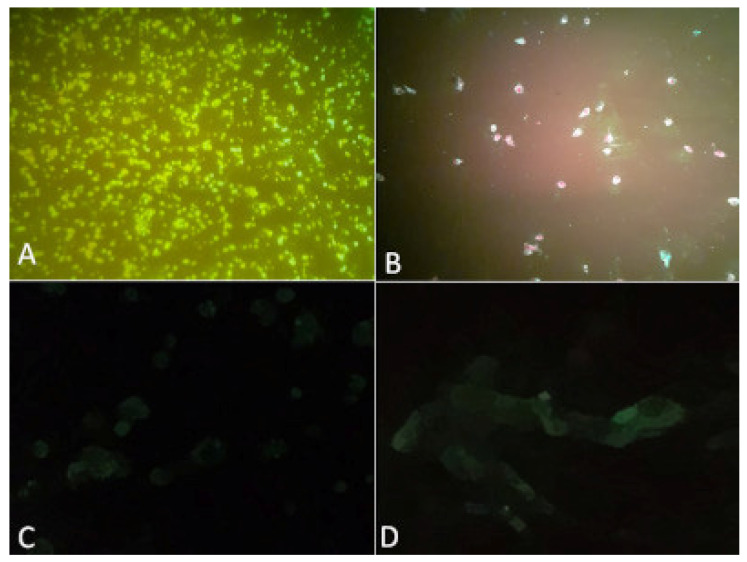
C6/36 and VERO cell cultures inoculated with the BeAR865640 sample, positive in the IFF test. (**A**) positive C6/36; (**B**) positive VERO; (**C**) C6/36 negative control; (**D**) VERO negative control.

**Figure 3 viruses-14-02101-f003:**
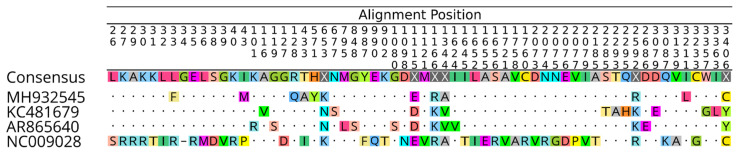
Alignment between the coding region of the isolate BeAr865640 and the coding regions of three complete ILHV sequences available at the NCBI.

**Figure 4 viruses-14-02101-f004:**
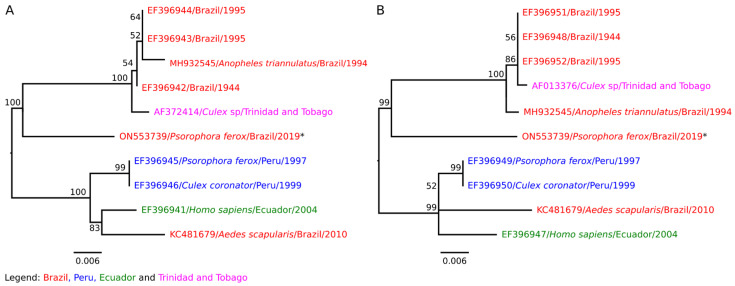
Phylogenetic relationship between the coding regions of the envelope proteins (**A**) and NS5 (**B**) of the isolate BeAr865640 (GenBank id. ON553739), with partial sequences referring to the coding regions of envelope proteins and NS5 of ILHV strains available at the NCBI. The Bootstrap values situated above branches represent the percentages derived from 1000 replicates, and the scale bar represents the nucleotide substitution rate. * Isolate BeAr865640.

**Table 1 viruses-14-02101-t001:** Panel of arboviruses used in the HI to detect antibodies in sera from free-ranging birds captured in APA-Belem, Pará, Brazil.

Family	Genus	Virus
*Togaviridae*	*Alphavirus*	*Eastern equine encephalitis virus* (EEEV) *Western equine encephalitis virus* (WEEV) *Mayaro virus* (MAYV) *Mucambo virus* (MUCV) *Pixuna virus* (PIXV) *Aura virus* (AURAV)
*Flaviviridae*	*Flavivirus*	*Saint Louis encephalitis virus* (SLEV) *West Nile virus* (WNV) *Yellow fever virus* (YFV) *Ilheus virus* (ILHV) *Cacipacore virus* (CPCV) *Bussuquara virus* (BSQV) *Rocio virus* (ROCV) *Naranjal virus* (NJLV)
*Peribunyaviridae*	*Orthobunyavirus*	*Tacaiuma orthobunyavirus* (TCMV) *Guaroa orthobunyavirus* (GROV) *Maguari orthobunyvirus* (MAGV) *Utinga orthobunyavirus* (UTIV) *Caraparu orthobunyavirus* (CARV) *Oropouche orthobunyavirus* (OROV) *Catu orthobunyavirus* (CATUV) *Marituba orthobunyavirus* (MTBV) *Murucutu orthobunyavirus* (MURV) *Oriboca orthobunyavirus* (ORIV) *Itaqui orthobunyavirus* (ITQV)
*Phenuiviridae*	*Phlebovirus*	*Icoaraci phlebovirus* (ICOV) *Bujaru phlebovirus* (BUJV) *Urucuri phlebovirus* (URUV)

**Table 2 viruses-14-02101-t002:** Polyclonal antibodies for antigenic groups of arboviruses used in the IFF test to detect viral presence in C6/36 and VERO cell cultures inoculated with supernatant of macerated hematophagous arthropods from APA-Belem, Pará, Brazil.

Genus	Antigenic Group	Virus
*Alphavirus*	A	AURAV, EEEV, MAYV, MUCV, PIXV, UNAV, WEEV, CHIKV, *Trocara virus*
*Flavivirus*	B	DENV 1–4, YFV, BSQV, CPCV, ILHV, NJLV, SLEV, WNV, ZIKAV
*Orthobunyavirus*	Guamá	*Ananindeua orthobunyavirus*, *Bimiti orthobunyavirus*, CATUV, *Guamá orthobunyavirus*, *Mirim orthobunyavirus*, *Moju orthobunyavirus*, *Timboteua orthobunyavirus.*
	Capim	*Acará orthobunyavirus*, *Benevides orthobunyavirus*, *Benfica orthobunyavirus*, *Capim orthobunyavirus*, *Guajará orthobunyavirus*, *Bushbush orthobunyavirus*, *Moriche orthobunyavirus.*
	Bunyamwera	*Iaco orthobunyavirus*, *Kairi orthobunyavirus*, *Macauã orthobunyavirus*, MAGV, *Sorocab orthobunyavirus*, *Tucunduba orthobunyavirus*, *Taiassui orthobunyavirus*, *Xingu orthobunyavirus.*
	Simbu	OROV
*Phlebovirus*	Phlebotomus	*Alenquer phlebovirus*, *Ambe phlebovirus*, *Ariquemes phlebovirus*, *Belterra phlebovirus*, BUJV, ICOV, *Candiru phlebovirus*, *Itaporanga phlebovirus*, *Jacundá phlebovirus*, *Joá phlebovirus*, *Morumbi phlebovirus*, *Mucura phlebovirus*, *Mugunba phlebovirus*, *Oriximiná phlebovirus*, *Pacuí phlebovirus*, *Salobo phlebovirus*, *Tapará phlebovirus*, *Turuna phlebovirus*, *Uriurana phlebovirus*, URUV.
*Orbivirus*	Changuinola	*Acatinga virus*, *Acurené virus*, *Almeirim virus*, *Altamira virus*, *Anapú virus*, *Araçaí virus*, *Aratau virus*, *Aruana virus*, *Arawetê virus*, *Assurinis virus*, *Bacajaí virus*, *Bacuri virus*, *Balbina virus*, *Barcarena virus*, *Breves virus*, *Canindé virus*, *Canoal virus*, *Catetê virus*, *Coari virus*, *Gorotire virus*, *Gurupi virus*, *Iopaka virus*, *Ipixaia virus*, *Irituia virus*, *Iruana virus*, *Itaboca virus*, *Jamanxi virus*, *Jandaia virus*, *Jari virus*, *Jatuarana virus*, *Jutaí virus*, *Kararaô virus*, *Melgaço virus*, *Monte Dourado virus*, *Ourém virus*, *Pacajá virus*, *Parakanã virus*, *Poranati virus*, *Parauapebas virus*, *Parú virus*, *Pependana virus*, *Pindobai virus*, *Piratuba virus*, *Purus virus*, *Rio Mutapi virus*, *Saracá virus*, *Serra Sul virus*, *Surubim virus*, *Tapiropé virus*, *Tekupeú virus*, *Timbozal virus*, *Tocantins virus*, *Tocaxá virus*, *Tuerê virus*, *Tumucumaque virus*, *Uatumã virus*, *Uxituba virus*, *Xaraíra virus*, *Xiwanga virus.*

**Table 3 viruses-14-02101-t003:** Free-ranging birds captured at APA-Belem (Pará, Brazil) were positive for the presence of arbovirus antibodies in the HI test.

Bird (Taxon)	Month/Year	Arbovirus/Title
Passeriformes		
Thraupidae		
*Ramphocelus carbo*	June/2019	ICOV/1:80
Thamnophilidae		
*Scalteria naevia*	July/2019	BUJV/1:80
*Taraba major*	March/2020	SLEV/1:20; NJLV/1:40
Dendrocolaptidae		
*Dendroplex picus*	March/2020	SLEV/1:20

**Table 4 viruses-14-02101-t004:** Nucleotide and amino acid identity matrix between the complete sequence of isolate BeAr865640 and three complete sequences of ILHV available on GenBank.

	Sequence (Id. GenBank)	1	2	3	4
1	MH932545		98.48%	99.4% ^2^	99.4%
2	NC_0093028	99.4%		98.5% ^2^	98.5%
3	ON553739 *	95.5% ^1^	95.2% ^1^		99.5% ^2^
4	KC481679	95.1%	94.7%	95.2% ^1^	

* Isolate BeAr865640; ^1^ Values corresponding to nucleotide similarity; ^2^ Values corresponding to amino acid similarity.

## Data Availability

Not applicable.

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
