# Peer review of "Arboviruses in Free-Ranging Birds and Hematophagous Arthropods (Diptera, Nematocera) from Forest Remnants and Urbanized Areas of an Environmental Protection Area in the Amazon Biome"

_viruses, 2022, doi:10.3390/v14102101_

Round 1

Reviewer 1 Report

1. There are no pictures of CPE and IIF to prove the results in Part 3.2 and the second sentence contains too many parentheses.

2. The expressions ofa high degree of identity between the amino acids and ”a lower degree of nucleotide identity” between lines 237-239 are different from Table 2, it may be legend errors according to your results,

3. In the 264th line, such should be such as.

4. In the 349th line, “ranges from 2 a 30%” should be “ranges from 2 to 30%‘’。

5. Is the word “impostant” in the 364th and 371st lines a writing error? Do you want to write it as “important”?

6. Are there any other findings besides isolate BeAr865640 to increase persuasiveness?

Author Response

1. Added photos that confirm the results of the IFF test.
Considerations 2., 3., 4. and 5. were corrected in the manuscript.
6. There are no further results to consider in this manuscript.

Reviewer 2 Report

In this study, the authors performed a survey of birds and arthropods collected from a study area in Belem, Brazil.  Collected bird sera were screened for hemagluttination inhibition using a panel of 29 arboviruses. Collected arthropods were macerated and added to cultured cells to detect replicating virus by indirect immunofluorescence.  A virus-positive sample was subjected to RNAseq and a sequence was obtained for Ilheus virus, which was compared to existing sequences for Ilheus virus in NCBI.

Overall this is a straightforward survey of arbovirus hosts and vectors in a study area that is a mix of urban and forested land in a large city in the Brazilian rainforest. It contributes to knowledge of arbovirus risks in this type of setting.

Comments:

 1. Line 45, the sentence “they remain infected for life, transmitting these viral agents to other arthropods through the sexual route, to their offspring through the transovarian route, and to vertebrate hosts through hematophagy” is rather misleading since it implies that all of these routes are used by all arboviruses. Evidence for sexual and transovarian transmission is lacking for most arboviruses.

2. Line 66, suggested wording change: “Cities like Belem, the capital of Para, have been spreading…”

3. Lines 64, 73-78, please remember that virus species are not biological entities, they are human-made intellectual concepts.  As such, a species cannot be isolated, or used in an assay, etc.  Instead, members of a species can be isolated, assayed, etc. Similarly in Figures 2 and 3, it is not possible to use a species in an assay, instead a virus isolate was used. Also correct this error in line 265 (Members of at least 31 different species have been detected…) and elsewhere.

4. What was the source of the viruses used in the HI assay? And the source of the polyclonal sera used in the IFF assay?

5. What is “human attraction protected and enlightened technique”?  Describe or provide a reference.

6. In Figure 2, it is not clear where the alphaviruses end and the flaviviruses begin in this table-like figure; same for flavi versus orthobunya.  Please clarify.  Similar comment for Fig. 3.

7. Please perform a spell check: Line 80, infection; Line 99, conducted; line 119, individuals; Line 152 appropriate; there are more errors but I stopped listing them at this point.

Author Response

All comments have been considered and corrected in the manuscript